# Sustainable Operation Structures in the Cross-Border Enterprise Ecosystem under Tariff Fluctuations: A Systematic Evaluation

**Ting Zeng** [1] and **Tianjian Yang** [2],*

1   School of Economics and Management, Beijing University of Posts and Telecommunications, Beijing 100876, China
2   School of Modern Post (School of Automation), Beijing University of Posts and Telecommunications, Beijing 100876, China
*   Correspondence: frankytj@bupt.edu.cn

**Abstract:** In a business ecosystem, cross-border enterprises are vulnerable to disruptions such as tariff fluctuations. By modeling distribution structures, this paper explores how cross-border enterprises develop ecological advantages and improve their resilience. With system dynamics methodology, a cross-border enterprise ecosystem is modeled in the context of employing a distributor in a foreign jurisdiction overseeing product distribution (as a commissionaire or a fully fledged distributor). This study compares the dynamic performance of different distribution structures in response to tariff changes. The comparative results reveal that enterprises with distributors are more resilient to tariff fluctuations than those without distributors. This study proposes an effective measure to mitigate the disruptions caused by a tariff change; reducing transfer prices within a range can help to recover some of the lost profits and sales caused by a tariff increase. Moreover, this research provides practical implications on ideal operating structures for various purposes under tariff changes; implementing a commissionaire model leads to the highest profit, while implementing a fully fledged distributor model provides a more friendly market environment to customers. The provided insights have theoretical and practical value for policy makers, managers and investors to deal with a wide spectrum of strategic business ecosystem challenges.

**Keywords:** business ecosystem; tariff fluctuation; distribution structure; system dynamics

## 1. Introduction

With increasing economic environment changes, how to build, maintain and expand competitive advantages has become a key issue for enterprises. The business ecosystem is based on heterogeneous enterprises and organizations which interact under mutual benefits [1]. Ecological advantages have two dimensions: overall strength and adaption to the external environment. Traditional advantages focus more on an independent enterprise's internal factors, while ecological advantages focus on the interaction between the enterprise and the business ecosystem. How enterprises consolidate and form new ecological advantages in the business ecosystem is explored from the perspective of cross-border businesses in this paper.

In recent years, tariffs have been fluctuating largely and frequently in a wide range of products trading worldwide. In July 2018, the United States imposed tariffs on USD 34 billion of Chinese goods, and China imposed retaliatory tariffs on US goods of a similar value [2,3]. In September 2019, China imposed 5 to 10% tariffs on one-third of the 5078 goods it imports from America, and the United States imposed new 15% tariffs on about USD 112 billion of Chinese imports [4]. Besides the China–United States trade conflict, tariff fluctuations are observed worldwide. Yet, little work can be found on the issues related to the impacts of tariff fluctuations on the business ecosystem or on the necessary framework

to deal with such challenges. This inspires our study of analyzing the resilience and sustainability of a cross-border business under tariff fluctuations.

When an exogenous disruption occurs, a firm tries to put its operation under control through internal capabilities. Nevertheless, risk management strategies designed proactively in advance with relevant partners are even more beneficial [5]. These strategies facilitate the continuous power of the enterprise over the external environment with the exchange of materials, energy and information between the enterprise and the environment. Since the disruption risk of tariff fluctuations is inescapable, this study seeks to solve the following research question: What can cross-border enterprises do to mitigate disruptions caused by a tariff increase? To resolve this problem, the dynamic performance of transfer prices is investigated through system dynamics modeling and simulation. The definition and utilities of transfer price have been widely demonstrated for the purpose of tax optimization [6–9]. This study attempts to reveal the utility of transfer price on developing ecological advantages.

This research addresses three main questions: (1) how cross-border enterprises are impacted by tariff changes; (2) how to mitigate disruptions caused by tariff fluctuations; and (3) how different operating structures contribute to developing ecological advantages. This paper makes four primary contributions. First, this study, for the first time, investigates the dynamic performance of cross-border enterprises impacted by tariff fluctuations through system dynamics modeling and simulation. The outcome shows that the relation between the change in tariff rate and sales is linear only if the price elasticity coefficient is equal to 1. Second, this study compares the performance of different distribution structures in response to a changing tariff. The comparative results reveal that enterprises with distributors in foreign markets are more resilient to a disruption stimulated by a tariff fluctuation. Third, this study proposes an effective measure to immediately mitigate the disruptions of tariff fluctuations on sales and profits. It suggests that reducing transfer prices within a range leads to a partial recovery of the decreased profits and sales induced by a tariff increase. Fourth, this study discusses the ideal operating structure under a tariff fluctuation for various purposes. It demonstrates that a commissionaire model is ideal for acquiring the highest profit and a fully fledged distributor model is ideal for establishing a friendly market environment for customer retention.

## 2. Literature Review

The related research is from two streams: sustainable business ecosystem and the impact of a tariff.

### 2.1. Sustainable Business Ecosystem

Moore [10] first proposed the definition of a business ecosystem as an economic community constituted by interacting organizations and individuals. Business ecosystems are based on core capabilities to produce core products. Gossain and Kandiah [11] saw a business ecosystem similar to an integrated value chain. The concept of a business ecosystem emphasizes relationships between organizations, the evolvement of those relationships and the significance of a brand. Lwein and Regine [12] defined a business ecosystem as a network of companies each occupying a place in its landscape of possibilities. Companies are coupled to competitors, collaborators and complementors. Changes in the landscape of a company cause changes in the landscapes of other participants of a business ecosystem due to interconnectedness. Power and Jerjian [13] claimed that one cannot manage a business without managing an entire ecosystem. There are four stakeholders to any enterprise: communities of shareholders, employees, customers and businesses. The key feature of an ecosystem is cooperation. A business ecosystem is built on one single company which is highly connected. Iansiti and Levien [14] described the business ecosystem as a network constituted by interconnected participants who depend on each other for mutual effectiveness and survival. In this ecosystem, participants share their fate together. There are three critical success factors of a business ecosystem: productivity, robustness and opportunities.

Peltoniemi and Vuori [15] defined a business ecosystem as a dynamic structure consisting of interconnected organizations. It develops through self-organization, emergence and co-evolution to achieve adaptability. Business ecosystems should be self-sustaining.

The sustainable business ecosystem is becoming an important research topic, especially in emerging economies. Sustainability has been studied from several perspectives such as flexibility, robustness, stability, dynamics, adaption and recovery [16]. Zhao et al. [17] studied how different network topologies affect the network's resilience against disruptions. Hasani et al. [18] presented a multi-objective optimization model to configure a global medical device manufacturing system under disruption. Hileman et al. [19] investigated sustainability challenges in a larger business ecosystem with the presence of organizations that occupy strategic positions around keystone actors. Limited research has focused on sustainability issues in the cross-border enterprise ecosystem.

*2.2. The Impact of Tariff*

Tariffs have been getting more attention worldwide in recent years, and the impact of a tariff is studied in various fields. The presence of tariffs influences not only the volume of imports but also the original decision concerning manufacturing locations in Horst's model [20]. The effects of tariff changes on overall welfare changes in the presence of foreign capital were studied by Bhagwati and Tironi [21]. Motta [22] analyzed the impact of a tariff and concluded that a tariff may induce a shift away from the foreign investment decision or lead to "tariff jumping" investment. Castellanos et al. [23] presented a techno-economic model with a tariff and transportation algorithm and demonstrated that introducing tariffs between countries significantly increases the minimum sustainable price for solar photovoltaics manufacturing and alters the optimal manufacturing locations. Giang [24] investigated the dynamic impact of a tariff on the economic growth in the Vietnamese economy between 1999 and 2017 empirically and found that tariffs have positive impacts on the economic growth. Barlow [25] studied the impact of tariff reduction and suggested that tariff changes have an under-recognized impact on public health expenditure and may contribute to global health spending disparities. Crowley et al. [26] investigated the impacts of tariff changes on foreign market entry decisions and suggested that by establishing stable tariff rates, trade agreements reduce one source of risk for firms to expand internationally.

Different from the foresaid research, this paper focuses on developing ecological advantages toward the risks caused by tariffs. This study employs system dynamics methodology as simulation models can describe complex problems and can be even more powerful than analytical closed-form analysis [27]. It presents a framework to evaluate the impact of tariff fluctuations on a cross-border enterprise ecosystem.

## 3. System Dynamics Modeling of a Cross-Border Enterprise

For cross-border enterprises with a sole manufacturing site, three commonly observed distribution operating structures are studied to evaluate the effect of tariff fluctuations: no distributor, commissionaire and fully fledged distributor [28,29] as outlined in Table 1. (1) No distributor: a cross-border enterprise with a headquarters (HQ), having no foreign distributor, undertakes all business actions and decisions for sales in the foreign jurisdiction. Products are shipped directly from HQ to foreign clients. The charged tariff amount is calculated based on the selling price to end customers. The optimal production volume is where HQ finds the maximization of after-tax profit ("overall profit"). (2) Commissionaire: a commissionaire only does sales and distribution with very limited authority and risk. HQ makes decisions for product distribution and profit allocation. In this case, the charged tariff amount is calculated based on the transfer price. The optimal purchase volume by the distributor is decided by HQ, where HQ finds the maximization of after-tax profit ("overall profit"). (3) Fully fledged distributor: a fully fledged distributor has the decision power and takes on business risks. It is a buy–sell entity that makes purchasing decisions and takes ownership of inventory. The charged tariff amount is calculated based on the transfer price.

The optimal purchase volume is hereby decided by the distributor itself, where it finds the maximization of after-tax profit ("distributor profit"). Besides these three structures, a limited-risk distribution structure was demonstrated to be rarely optimal [29], and thus it is not considered in this study.

**Table 1.** Characteristics of different distribution structures.

|  | No Distributor | Commissionaire | Fully Fledged Distributor |
|---|---|---|---|
| Manufacturing at HQ? | Yes | Yes | Yes |
| Foreign Distributor Implemented? | No | Yes | Yes |
| Distributor with Decision Power? | No | No | Yes |

System dynamics has been shown to be a well-suited modeling and analysis tool for strategic management [30–32]. We model a cross-border enterprise using the notation summarized in Table 2.

**Table 2.** Nomenclature list and notation.

| Nomenclature | Notation | Description | Units |
|---|---|---|---|
| **Level Variable** | | | |
| HQ's inventory | $I_s$ | The accumulative product volume that is in stock for shipping | Unit |
| Distributor's bonded inventory | $I_b$ | Product volume in bonded inventory | Unit |
| Distributor's unbonded inventory | $I_u$ | Product volume in unbonded inventory | Unit |
| **Rate Variable** | | | |
| Sales | $V_s$ | Sales rate | Unit/month |
| Target production volume | $V^*$ | Optimal production rate | Unit/month |
| **Auxiliary Variable** | | | |
| Sales price w tariff | $p_w$ | Sales price with tariff | USD/unit |
| Sales price w/o tariff | $p_{w/o}$ | Sales price without tariff | USD/unit |
| Transfer coefficient | $\mu$ | A percentage that is equal to transfer price divided by sales price | Dimensionless |
| Transfer price | $p_d$ | Transfer price charged to distributor | USD/unit |
| **Exogenous Variable** | | | |
| Distributor local tax rate | $X_s$ | Ad valorem sales tax rate [33] including all local tax | Dimensionless |
| HQ export tax rate | $X_d$ | Ad valorem export tax rate in HQ region [33] | Dimensionless |
| Non-linear demand coefficient | $\rho$ | The coefficient in a non-linear demand function | Dimensionless |
| Unit production cost | $c$ | Production cost per unit product | USD/unit |
| Sales unit cost | $c_o$ | The cost for selling one unit of product, including packaging cost, local delivery cost, delivery insurance | USD/unit |
| Shipment cost | $s$ | The average monthly cost for routine shipping | USD/month |
| Shipment cost per order | $s_0$ | The average cost for shipping an order | USD/unit |
| Tariff rate | $X_c$ | Ad valorem import tariff rate in distributor region | Dimensionless |

To make the analysis tractable, the following assumptions are made in this research:

**Assumption 1.** *Enterprises and distributors, who are rational decision makers and risk neutral, are pursuing profit maximization. For each unit sold, marginal profit is obtained from marginal revenue (MR) minus marginal cost (MC). At the output level where marginal revenue equals marginal cost, the profit is maximized [34]. If marginal revenue is still greater than marginal cost, enterprises are inclined to produce a greater quantity, and vice versa.*

**Assumption 2.** *The market demand Q is affected by price p with an elasticity e [35,36]. The elasticity of demand is assumed as*

$$e = -\frac{dQ}{dp} * \frac{p}{Q} \tag{1}$$

The demand function is assumed non-linear as

$$Q = \rho * p^{-e} \tag{2}$$

where $\rho$ is a coefficient and $e > 1$. As most branded natural honey is manufactured in a sole site and sold cross-border, we take the honey industry as an example that matches our model assumptions. On the basis of the year 2010's honey production data [37], we derive approximately $e = 6$ and $\rho = 5 \times 10^{14}$.

*3.1. Causal Loop Diagrams*

On the basis of an operation process of a cross-border business, causal loop diagrams are constructed depicting three distribution structures: no distributor, commissionaire and fully fledged distributor.

3.1.1. No Distributor Model

In Figure 1, we present a model with no distributor in its causal loop diagram. For a cross-border enterprise that has no distributor, it has a headquarters (HQ) in its home jurisdiction and sells directly into a foreign jurisdiction. The enterprise pays for export tax and customers pay for tariff tax. Given assumption 1, HQ determines the target production volume, which maximizes its profit. The flow of products goes through the stages of *Production*, *Inventory*, *Transportation* and *Sales*. The *Tariff* rate and *Export Tax* influence both the *Cost* and *Price*. The activities of *Production*, *Transportation* and *Sales* generate *Cost*. The *Production* rate is determined by the *Target Production Volume*. To achieve maximum profit, HQ employs the optimal sales volume as the *Target Production Volume* and aims to sell out all manufactured products. Thus, the *Target Production Volume* is determined when marginal *Revenue* equals marginal *Cost*. The *Sales* rate is determined by *Sales Effort*, *Price* and *Transportation*. We assume a market-clearing mechanism [29], which indicates the inventory equals the demand. Given this assumption, *Price* is determined by *Inventory*.

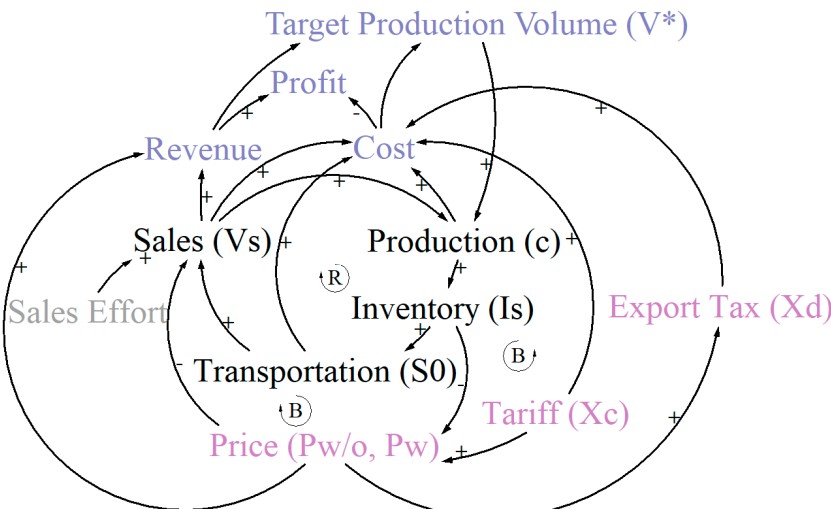

**Figure 1.** Causal loop diagram of a no-distributor model.

Total *Cost* is a sum of production cost, operation cost, transportation cost, sales cost and taxes. As a minor change in product volume would not have an evident impact on operation cost and sales cost, operation cost and sales cost are considered to be fixed and weakly bonded to units of products. HQ manufactures products at a cost $c$ per unit product and delivers products at a unit cost of $s_0$ per order. For simplicity, we assume one order contains only one unit of product, and thus the number of orders equals the number of ordered products. HQ pays export tax at a rate of $X_d$ with a selling price $p_{w/o}$,

and customers pay tariff tax at a rate of $X_c$ with a purchasing price $p_w$. When marginal revenue equals marginal cost, HQ achieves the highest profit.

**Proposition 1.** *In a no-distributor model, the optimal sales price and sales volume for HQ are given as:*

$$p_{w/o} = \frac{c + s_0}{(1 - X_d)\left(1 - \frac{1}{e}\right)} \tag{3}$$

$$V^* = V_s = \rho * \left(\frac{(1 - X_d)\left(1 - \frac{1}{e}\right)}{(1 + X_c)(c + s_0)}\right)^e \tag{4}$$

For proof see Appendix A.

The optimal sales volume per month ($V^*$) is employed by HQ as the target production volume per month to monitor production. The production rate is determined by target production volume and production capacity.

### 3.1.2. Commissionaire Model

Figure 2 presents a causal loop diagram of a "commissionaire" model, in which the cross-border enterprise has a commissionaire distributor in a foreign jurisdiction. A commissionaire distributor only does sales and distribution, while HQ makes centralized decisions based on profit maximization. In spite of adding overhead costs, the insertion of a distribution division adds overhead costs, but it helps to further penetrate the market with local sales efforts and a shorter delivery timeline to customers. Unlike the "no-distributor" structure, HQ does not do sales. The local distributor is employed to manage inventories and conduct sales activities. A *Transfer Price* is charged by HQ to the distributor for each unit of products. In this model, *Price* is determined by HQ's *Inventory* ($I_s$), distributor's *Bonded Inventory* ($I_b$) and *Unbonded Inventory* ($I_u$).

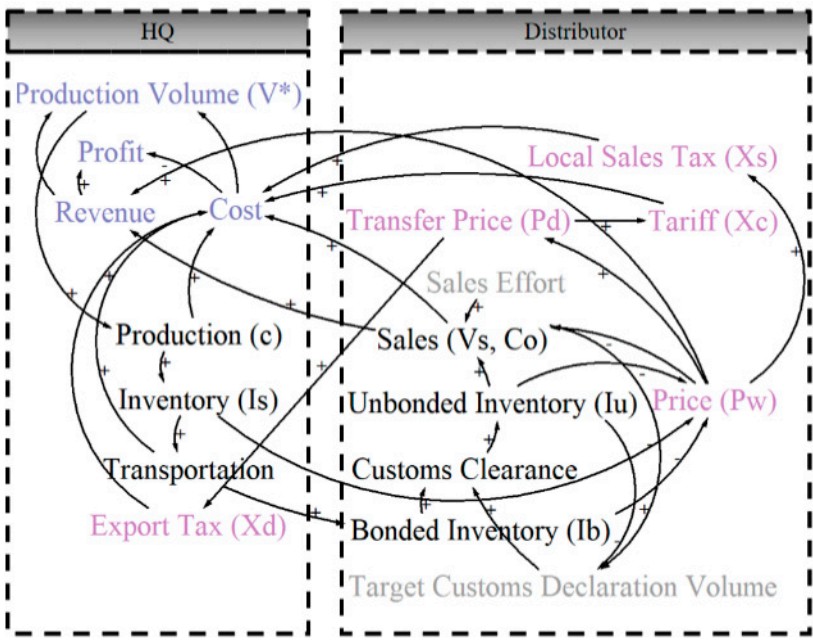

**Figure 2.** Causal loop diagram of a commissionaire model.

A distributor is employed as a commissionaire, and HQ determines the target production volume according to the profit maximization concept (i.e., marginal revenue equals marginal cost). Total *Cost* is a sum of production cost, operation cost, transportation cost, sales cost and taxes. Among these costs, production cost $c$, unit sales cost $c_o$ (for packaging,

local delivery, delivery insurance, etc.) and taxes are strongly bonded to units of products. HQ ships products routinely to the distributor warehouse with a lower frequency than that in the "no-distributor" model to control shipping cost. A transfer price $p_d$ is what the distributor pays to HQ for each product. When marginal revenue equals marginal cost, the firm achieves the highest profit.

**Proposition 2.** *In a commissionaire model, the optimal sales price and sales volume for HQ are obtained as:*

$$p_w = \frac{c + c_o + p_d * X_d + p_d * X_c}{(1 - X_s) * \left(1 - \frac{1}{e}\right)} \tag{5}$$

$$V^* = V_s = \rho * \left( \frac{(1 - X_s) * \left(1 - \frac{1}{e}\right)}{c + c_o + p_d * X_d + p_d * X_c} \right)^e \tag{6}$$

For proof see Appendix B.

In the structure of a commissionaire model, the distributor pays for tariffs when products get through the customs. Eventually, the tariff cost is transferred to customers when they purchase products from the distributor. The sales effort in the model of a commissionaire is considered higher than that in the "no-distributor" model due to the existence of a local distributor. As the distributor is a commissionaire, transfer price is determined by HQ, and it reflects how much profit HQ transfers to the distributor.

### 3.1.3. Fully Fledged Distributor Model

Different from a commissionaire, a fully fledged distributor operates independently, makes the purchase decisions and bears business risks. HQ usually establishes a contract with the distributor by offering either a constant transfer coefficient (i.e., a fixed discount) or a fixed transfer price. With a constant transfer coefficient, HQ and the distributor share risks and revenue. With a fixed transfer price, the distributor takes all the risks. The *Order Volume* is determined by the distributor's marginal *Revenue* and marginal *Cost*. The *Production* rate is determined by the distributor's *Order Volume*. The sole difference between Figure 3a,b is whether there exists a connection between *Price* and *Transfer Price*. As a fully fledged distributor has a higher motivation to sell products than in previous structures, the sales effort is considered higher than the structure of "no distributor" or "commissionaire".

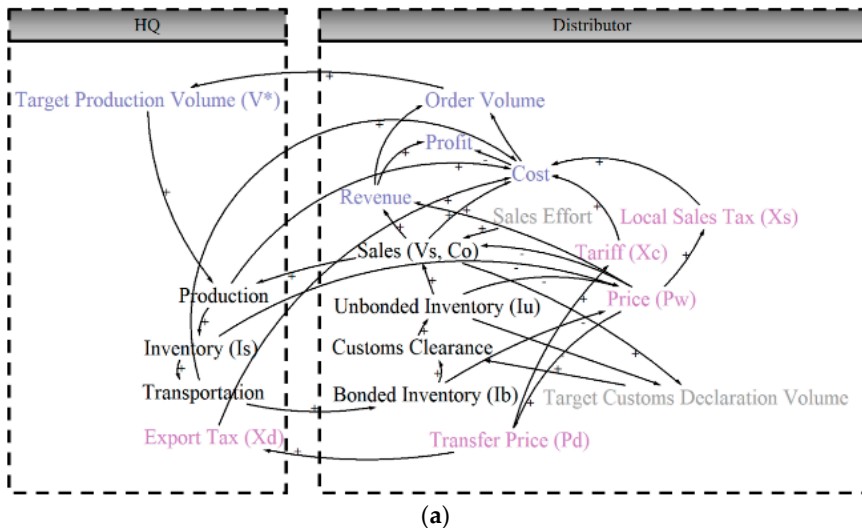

**(a)**

**Figure 3.** *Cont.*

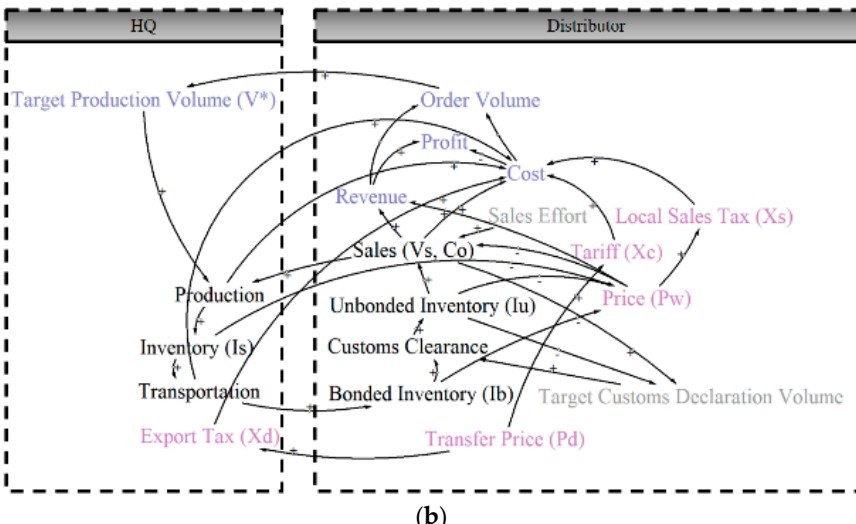

**(b)**

**Figure 3.** Causal loop diagrams of a fully fledged distributor model. (**a**) A fixed transfer coefficient. (**b**) A fixed transfer price.

In Figure 3a, transfer price $p_d$ changes along with the sales price $p_w$ by a transfer coefficient $\mu$ as $p_d = p_w * \mu$. In Figure 3b, transfer price remains constant although the sales price changes. In a fully fledged distributor model (either with a constant transfer coefficient or a fixed transfer price), the order volume and target production volume are determined by distributor revenue and cost.

**Proposition 3.** *In a fully fledged model, the optimal sales price and order volume for the distributor are given as:*

$$p_w = \frac{c_o + p_d + p_d * X_c}{(1 - X_s)\left(1 - \frac{1}{e}\right)} \tag{7}$$

$$V^* = V_s = \rho * \left(\frac{(1 - X_s) * \left(1 - \frac{1}{e}\right)}{c_o + p_d + p_d * X_c}\right)^e \tag{8}$$

For proof see Appendix C.

*3.2. Simulation Models*

On the basis of causal loop diagrams, the stock–flow diagrams for three distribution structures (i.e., no distributor, commissionaire and fully fledged distributor) are developed on the *Vensim*® (Ventana Systems, Inc., Harvard, MA, USA) platform as presented in Figure 4. The embedded mathematical equations of the models are available from the authors upon request.

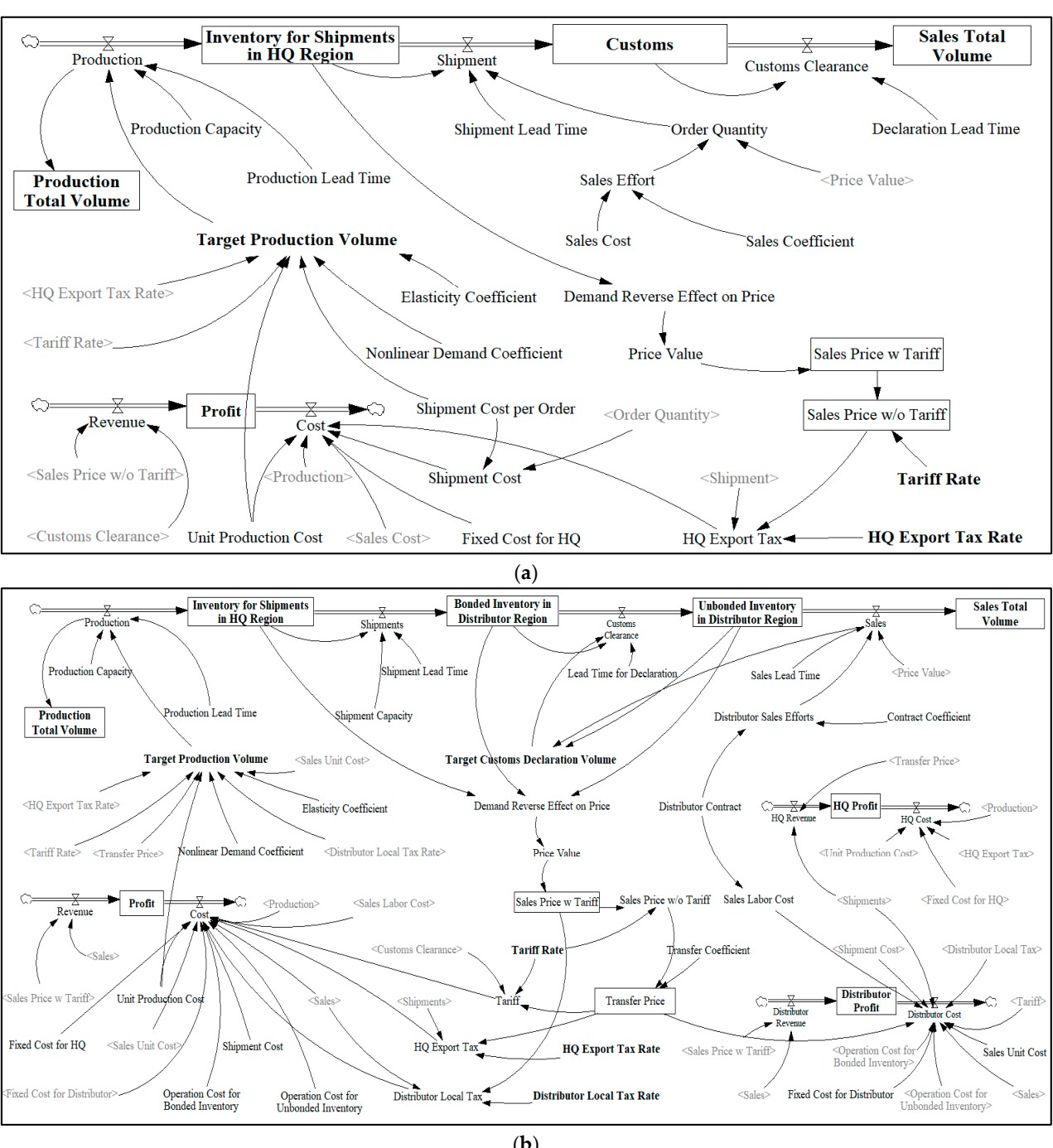

(**a**)

(**b**)

**Figure 4.** *Cont*.

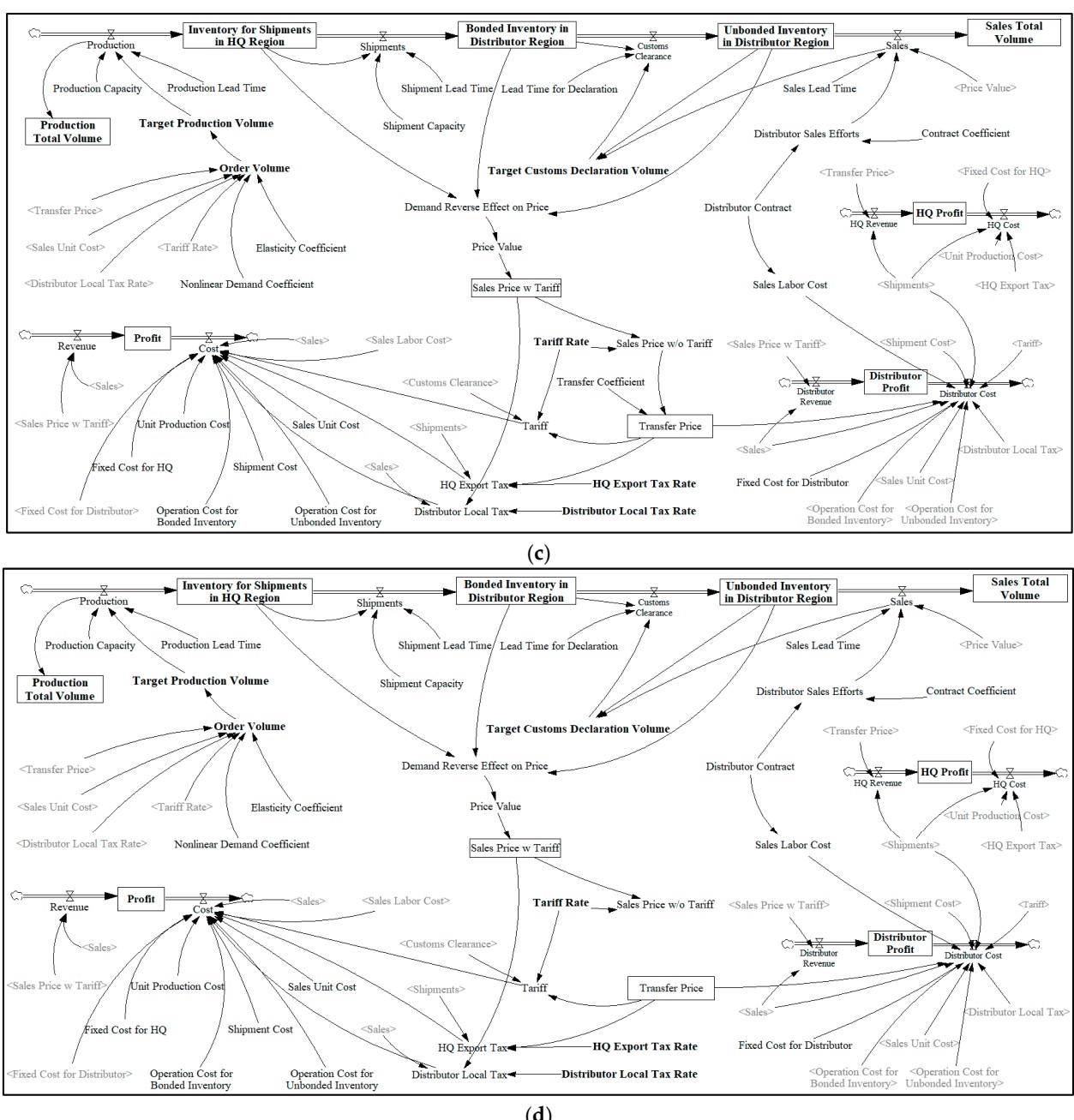

**Figure 4.** The stock–flow diagram of the system dynamics model under the operation structure with (**a**) no distributor, (**b**) a commissionaire, (**c**) a fully fledged distributor at a constant transfer coefficient, and (**d**) a fully fledged distributor at a fixed transfer price.

## 4. Simulation Results

Several graphical simulation programs (e.g., *i-think*®, *Powersim*®, *Stella*®, *Vensim*®) support the study of dynamic systems. In this paper, *Vensim*® is used for simulation. The natural honey tax data, extracted from the World Trade Organization Tariff Download Facility, shows 43 out of 128 countries have an ad valorem tariff rate ranging from 10 to 30%. Accordingly, we simulate the models with tariff rates at 10%, 20% and 30%, respectively.

### 4.1. No-Distributor Model Simulation

The computational results in Figure 5 show that, as the tariff rate increases, both profit and sales volume decrease; however, the impact of the tariff rate on both profit and sales volume becomes weaker as the tariff rate goes further up. From Equation (4), it can be

seen that the target production volume is affected by the tariff rate with an exponent $e$, which is a price elasticity coefficient. When $e = 1$, the impact of the tariff on profit is linear; otherwise, it is non-linear.

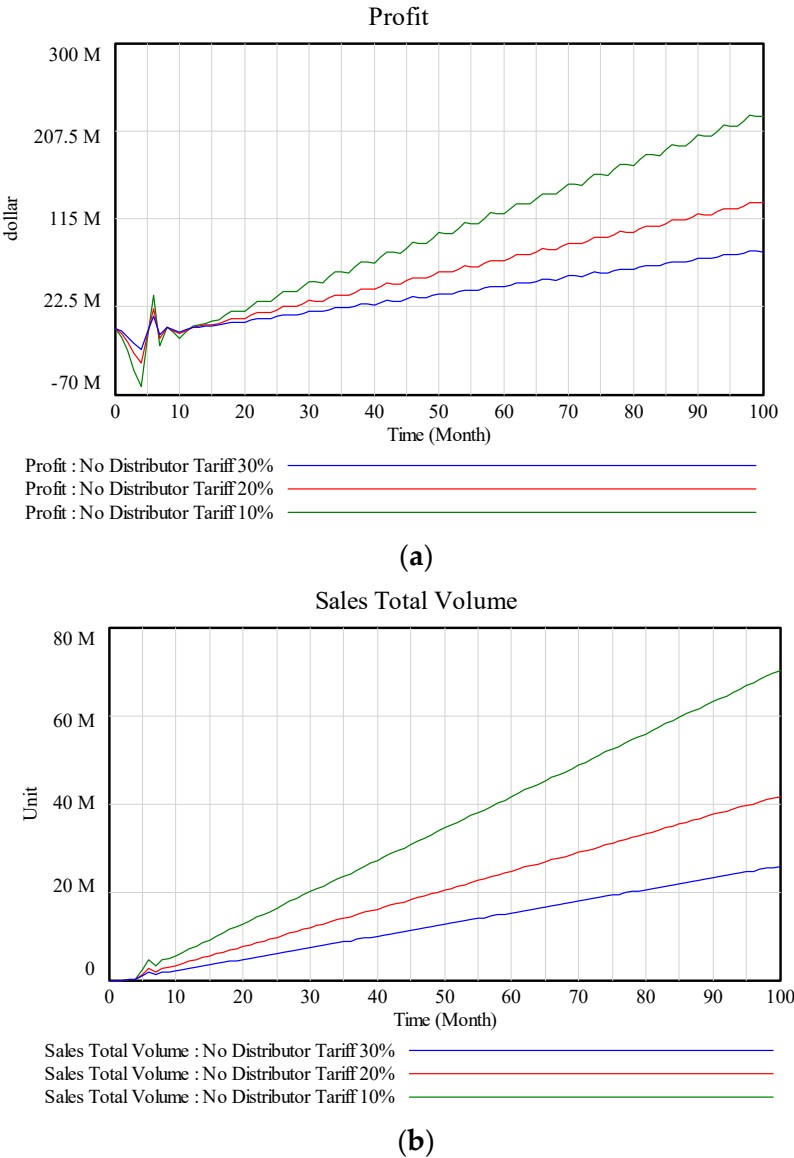

**Figure 5.** Simulation results of no-distributor model: (**a**) profit, (**b**) sales volume.

### 4.2. Commissionaire Model Simulation

The simulation results of a commissionaire model show the same pattern as that of a "no-distributor" model: when the tariff rate increases, profits and sales volume decrease while the impact of the tariff gets weaker. To reduce the economic loss caused by an increase in tariff rate, the enterprise adjusts the transfer coefficient (i.e., discount) to transfer more profit to the distributor. As shown in Figure 6, profit is partially recovered by decreasing the transfer coefficient from 60 to 40%. The adjustment of the transfer coefficient is an effective measure in response to an increasing tariff rate. However, due to the arm's length principle invented by tax administrations, the transfer coefficient should stay in a fair range. The arm's length principle indicates that related entities should agree on the same terms and conditions which would have been agreed upon between non-related entities for comparable uncontrolled transactions. Therefore, transfer price should be adjusted within a certain range to strategically reduce the impact of tariff fluctuations.

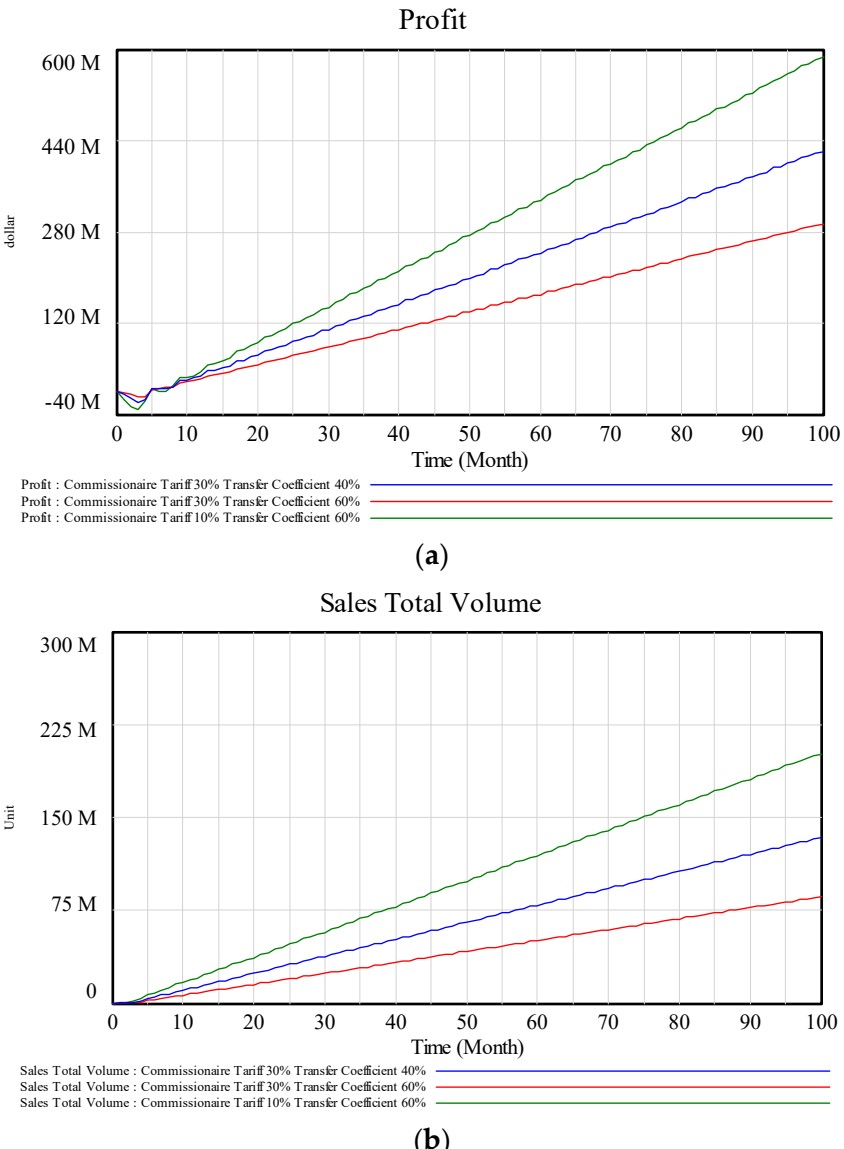

**Figure 6.** Simulation results of commissionaire model with an adjustment of transfer coefficient: (**a**) profit, (**b**) sales volume.

### 4.3. Fully Fledged Distributor Model Simulation

4.3.1. Fixed Transfer Coefficient

The computational results of a fully fledged distributor model with a constant transfer coefficient demonstrate that the total sales volume remains the same under a tariff fluctuation, as shown in Figure 7. The distributor profit shows no obvious change under the impact of tariff fluctuations.

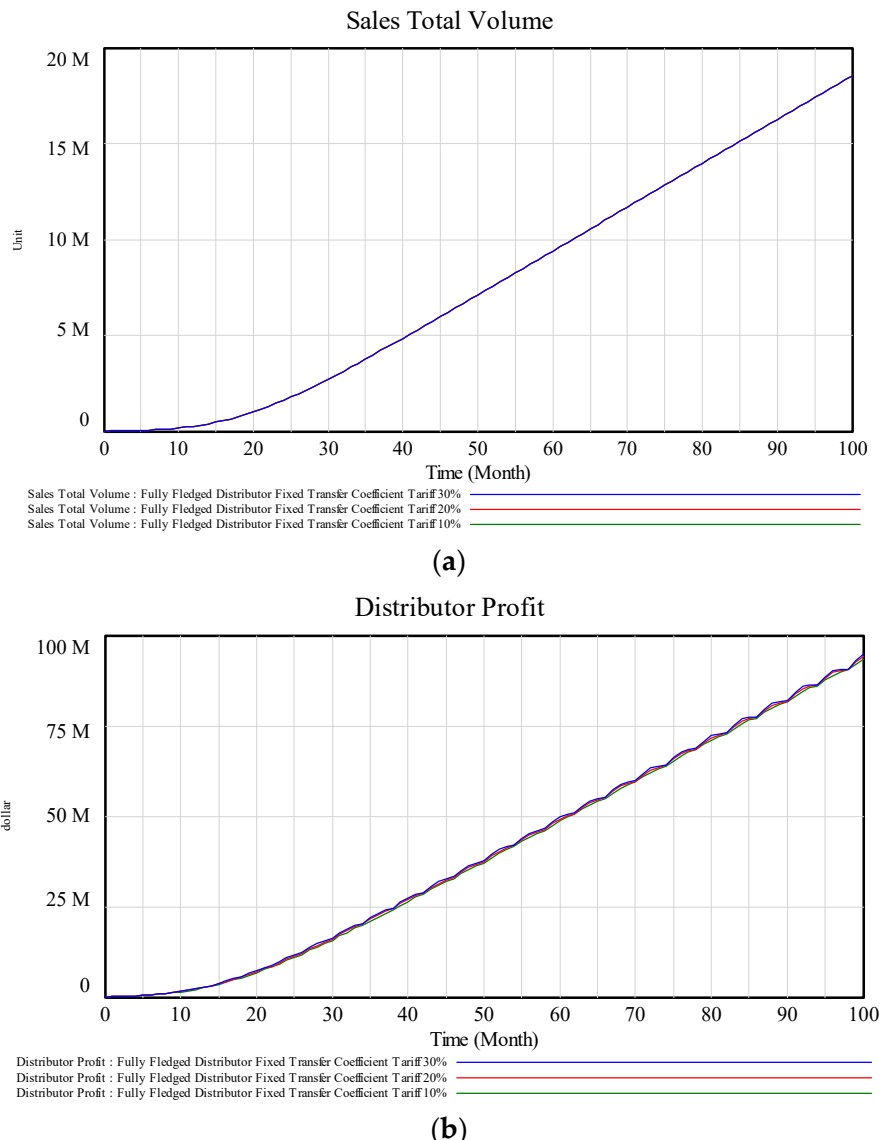

**Figure 7.** Simulation results of a fully fledged distributor model with a fixed transfer coefficient: (**a**) sales volume, (**b**) distributor profit.

### 4.3.2. Fixed Transfer Price

Figure 8 shows the simulation results of a fully fledged distributor model with a fixed transfer price. The profit fluctuates even with a stable tariff rate. If the tariff rate fluctuates, the profit oscillates even more widely.

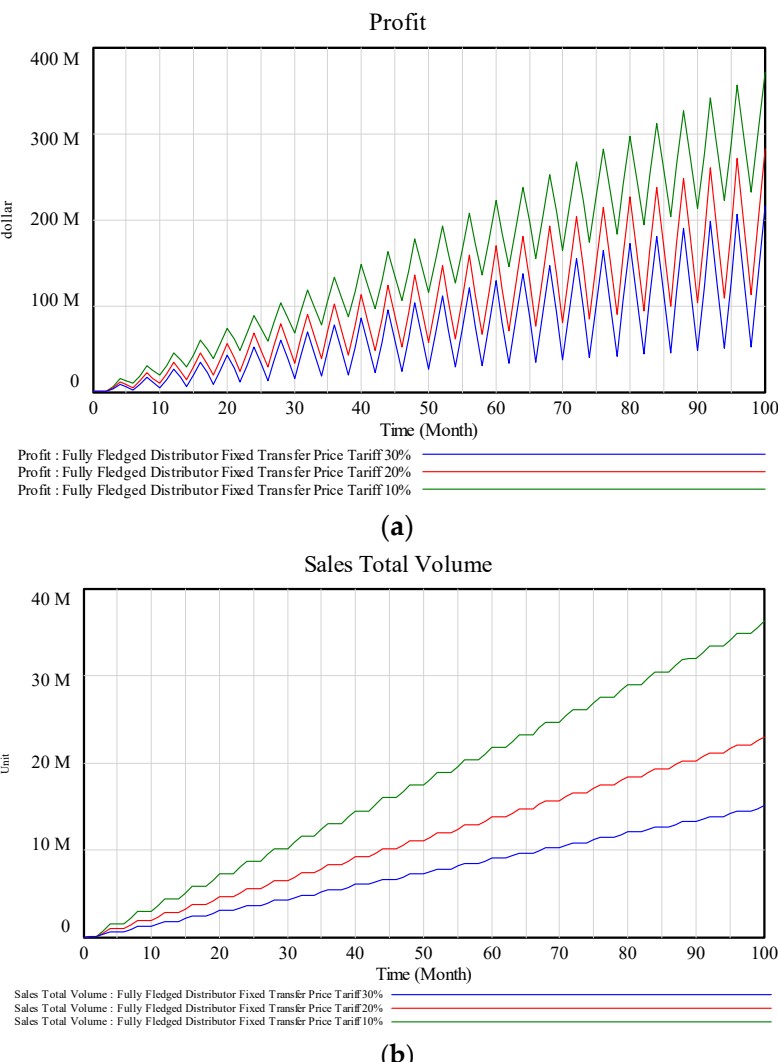

**Figure 8.** Simulation results of a fully fledged distributor model with a fixed transfer price: (**a**) profit, (**b**) sales volume.

## 5. Comparison and Discussion of Different Structures

A summary of the simulation results of different structures is presented in Table 3. A transfer price is determined when a distributor is employed. The total sales volume of a fully fledged distributor with a fixed transfer coefficient is not impacted by a tariff fluctuation, while that of other structures is affected. The overall profit and HQ profit of all structures show similar patterns under a tariff fluctuation. However, the tariff fluctuation has different impacts on the distributor profit of a commissionaire and a fully fledged distributor. The sales price of a fully fledged distributor with a fixed transfer coefficient remains the same under tariff fluctuations. With other structures, the sales price increases as the tariff rate increases. Hence, the distribution structure of a fully fledged distributor with a fixed transfer coefficient is most friendly to customers as the sales price is stable.

**Table 3.** Simulation results of different structures under tariff fluctuations.

| Measure | No Distributor | Commissionaire | Fully Fledged Distributor | |
|---|---|---|---|---|
| | | | **Fixed Transfer Coefficient** | **Fixed Transfer Price** |
| Optimal Sales Volume | $\rho * \left( \frac{\left(1-X_d\right)\left(1-\frac{1}{e}\right)}{\left(1+X_c\right)\left(c+s_0\right)} \right)^e$ | $\rho * \left( \frac{\left(1-X_s\right)*\left(1-\frac{1}{e}\right)}{c+p_d*X_d+p_d*X_c} \right)^e$ | $\rho * \left( \frac{\left(1-X_s\right)*\left(1-\frac{1}{e}\right)}{c_o+p_d+p_d*X_c} \right)^e$ | $\rho * \left( \frac{\left(1-X_s\right)*\left(1-\frac{1}{e}\right)}{c_o+p_d+p_d*X_c} \right)^e$ |
| Transfer Price | N/A | Adjustable | Adjustable | Constant |
| Total Sales Volume | Affected by tariffs | Affected by tariffs; recovered by adjusting transfer price | Not affected by tariffs | Affected by tariffs |
| Overall Profit | Negatively impacted by tariffs | Negatively impacted by tariffs | Negatively impacted by tariffs | Negatively impacted by tariffs |
| HQ Profit | N/A | Negatively impacted by tariffs | Negatively impacted by tariffs | Negatively impacted by tariffs |
| Distributor Profit | N/A | Negatively impacted by tariffs | Slightly positively impacted by tariffs | Slightly negatively impacted by tariffs |

A comparison of the overall profit for each structure is presented in Figure 9. It shows that a cross-border enterprise acquires the highest profit when it employs a commissionaire structure (red line), rather than other structures.

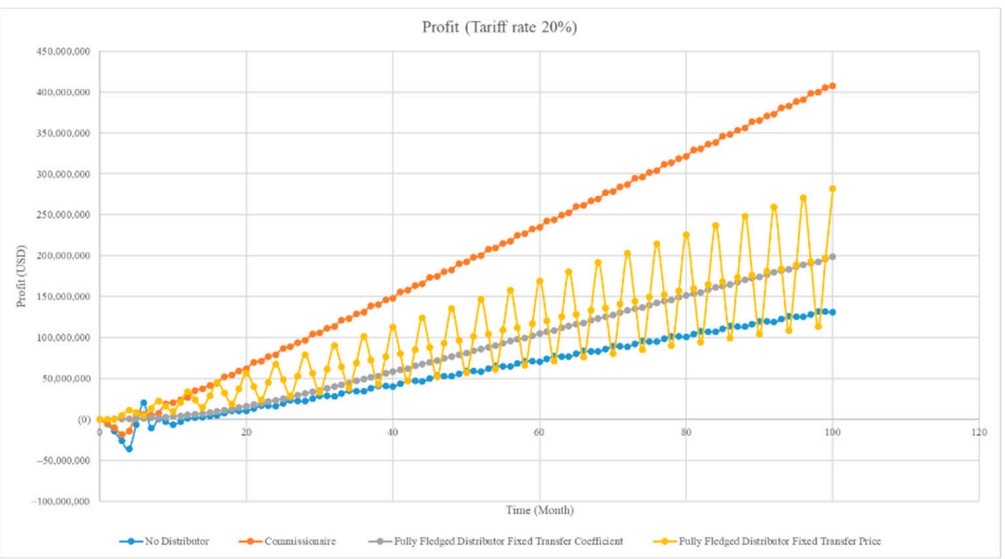

**Figure 9.** Profit with different structures when tariff rate is at 20%.

## 6. Conclusions and Future Research

To achieve long-term sustainable development, cross-border enterprises need to strengthen their capabilities by considering the health status of their entire ecosystem. By weighing in on which of the structural options a cross-border enterprise should implement, we help enterprises choose an appropriate way of enhancing ecological advantages in the business ecosystem under disruptions. No distributor, commissionaire and fully fledged distributor are possible ways to structure distribution operations of a cross-border enterprise. This research constructed modeling of an enterprise ecosystem including multiple decision makers, flows of products, profit optimization and interactions among enterprises, distributors, customers and governments.

This paper investigates business ecosystem sustainability in three steps: assessing possible structures, constructing dynamic models and developing ecological advantages. The progression of our analysis highlighted each advantage and disadvantage that occurs in the operation of each structure. In summary, this paper makes four primary contributions. First, this study is the first to investigate the dynamic performance of cross-border enterprises impacted by tariff fluctuations through system dynamics modeling and simulation. The outcome shows that the relation between tariff rate and sales is linear only if the price elasticity coefficient is equal to 1. Second, this study compares the performance of different distribution structures in response to a changing tariff. The comparative results reveal

that cross-border enterprises with distributors are more resilient to tariff fluctuations than those without distributors. Third, this study proposes an effective measure to immediately mitigate the disruptions of tariff fluctuations on sales and profits. It suggests that reducing transfer prices within a range leads to a partial recovery of the decreased profits and sales induced by a tariff increase. Fourth, this study discusses the ideal operating structure for various purposes under a tariff fluctuation. It demonstrates that a commissionaire model is ideal for acquiring the highest profit, while a fully fledged distributor model is ideal for establishing a friendly market environment for customer retention. This study provides full comparative statistics for the decisions and financial outcomes induced by each of the operating structures. The structural optimization of enterprises promotes the development of business ecosystem sustainability.

This research draws several conclusions with theoretical and practical value. The developed models may prove useful to policy makers and decision makers who are dealing with a wide spectrum of strategic business ecosystem issues. However, the enterprise ecosystem in real life is constituted of a large number of components and is more complicated than constructed models. In future studies, we will expand the dynamic models to include more elements of the business ecosystem and analyze various scenarios under different conditions.

**Supplementary Materials:** The following supporting information can be downloaded at: https://www.mdpi.com/article/10.3390/systems10060211/s1.

**Author Contributions:** Conceptualization and methodology, T.Z. and T.Y.; data curation, formal analysis, investigation, software, visualization and writing—original draft, T.Z.; funding acquisition, project administration, resources, supervision, validation and writing—review and editing, T.Y. All authors have read and agreed to the published version of the manuscript.

**Funding:** This research is funded by the National Natural Science Foundation of China, grant number 71001010.

**Data Availability Statement:** Data supporting the findings of this study were generated using the equations listed in the supplementary material and are also available from authors on request.

**Conflicts of Interest:** The authors declare no conflict of interest.

**Appendix A**

**Proof of Proposition A1.** Given Equation (2) and the market clearing mechanism, we have

$$e = -\frac{dI_s}{dp_{w/o}} * \frac{p_{w/o}}{I_s}$$

By the definition of marginal revenue, it can be observed that

$$MR = \frac{dR}{dI_s} = \frac{d(p_{w/o} * I_s)}{dI_s} = p_{w/o} * \left(\frac{dI_s}{dI_s}\right) + \left(\frac{dp_{w/o}}{dI_s}\right) * I_s = p_{w/o} + p_{w/o} * \left(\frac{dp_{w/o}}{dI_s}\right) * \left(\frac{I_s}{p_{w/o}}\right)$$
$$= p_{w/o} * \left[1 + \left(\frac{dp_{w/o}}{dI_s}\right) * \left(\frac{I_s}{p_{w/o}}\right)\right]$$

By substituting $e$ into $MR$, we get

$$MR = (1 - \frac{1}{e})p_{w/o}$$

By the definition of marginal cost, it can be found that

$$MC = c + s_0 + X_d * MR = c + s_0 + X_d * p_{w/o} * (1 - \frac{1}{e})$$

Given assumption 1 of profit maximization when $MR = MC$, it can be concluded that

$$(1 - \frac{1}{e})p_{w/o} = c + s_0 + X_d * p_{w/o} * (1 - \frac{1}{e})$$

By simplifying the equation, we arrive at

$$p_{w/o} = \frac{c + s_0}{(1 - X_d)\left(1 - \frac{1}{e}\right)}$$

By combining the above equation, Equation (2) and the definition of $p_w = p_{w/o} * (1 + X_c)$, the optimal sales volume can be determined as

$$V^* = V_s = \rho * (\frac{(1 - X_d)(1 - \frac{1}{e})}{(1 + X_c)(c + s_0)})^e$$

$\square$

**Appendix B**

**Proof of Proposition A2.** Given Equation (2) and the market clearing mechanism, we have

$$e = -\frac{d(I_s + I_b + I_u)}{dp_w} * \frac{p_w}{I_s + I_b + I_u}$$

By the definition of marginal revenue, it can be observed that

$$MR = \frac{dR}{d(I_s+I_b+I_u)} = \frac{d(p_w*(I_s+I_b+I_u))}{d(I_s+I_b+I_u)} = p_w * \left(\frac{d(I_s+I_b+I_u)}{d(I_s+I_b+I_u)}\right) + \left(\frac{dp_w}{d(I_s+I_b+I_u)}\right) * (I_s + I_b + I_u) = p_w + p_w * \left(\frac{dp_w}{d(I_s+I_b+I_u)}\right) * \left(\frac{I_s+I_b+I_u}{p_w}\right) = p_w * \left[1 + \left(\frac{dp_w}{d(I_s+I_b+I_u)}\right) * \left(\frac{I_s+I_b+I_u}{p_w}\right)\right]$$

By substituting $e$ into $MR$, we get

$$MR = (1 - \frac{1}{e})p_w$$

By the definition of marginal cost, it can be found that

$$MC = c + c_o + X_d * p_d + X_c * p_d + X_s * MR = c + c_o + X_d * p_d + X_c * p_d + X_s * p_w * (1 - \frac{1}{e})$$

Given assumption 1 of profit maximization when $MR = MC$, we arrive at

$$p_w = \frac{c + c_o + p_d * X_d + p_d * X_c}{(1 - X_s) * \left(1 - \frac{1}{e}\right)}$$

Given Equation (2), the optimal sales volume can be determined as

$$V^* = V_s = \rho * (\frac{(1 - X_s) * (1 - \frac{1}{e})}{c + c_o + p_d * X_d + p_d * X_c})^e$$

$\square$

**Appendix C**

**Proof of Proposition A3.** Given Equation (2) and the market clearing mechanism, we have

$$e = -\frac{d(I_s + I_b + I_u)}{dp_w} * \frac{p_w}{I_s + I_b + I_u}$$

By the definition of marginal revenue, it can be observed that

$$MR = \frac{dR}{d(I_s+I_b+I_u)} = \frac{d(p_w*(I_s+I_b+I_u))}{d(I_s+I_b+I_u)} = p_w * \left(\frac{d(I_s+I_b+I_u)}{d(I_s+I_b+I_u)}\right) + \left(\frac{dp_w}{d(I_s+I_b+I_u)}\right) * (I_s + I_b + I_u) = p_w + p_w * \left(\frac{dp_w}{d(I_s+I_b+I_u)}\right) * \left(\frac{I_s+I_b+I_u}{p_w}\right) = p_w * \left[1 + \left(\frac{dp_w}{d(I_s+I_b+I_u)}\right) * \left(\frac{I_s+I_b+I_u}{p_w}\right)\right]$$

By substituting $e$ into $MR$, we get

$$MR = (1 - \frac{1}{e})p_w$$

By the definition of marginal cost, it can be concluded that

$$MC = p_d + c_o + X_c * p_d + X_s * MR = p_d + c_o + X_c * p_d + X_s * p_w * (1 - \frac{1}{e})$$

Given assumption 1 of profit maximization when we have $MR = MC$, we arrive at

$$p_w = \frac{c_o + p_d + p_d * X_c}{(1 - X_s)\left(1 - \frac{1}{e}\right)}$$

Given Equation (2), the optimal sales volume can be determined as

$$V^* = V_s = \rho * \left(\frac{(1 - X_s) * (1 - \frac{1}{e})}{c_o + p_d + p_d * X_c}\right)^e$$

$\square$

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
