# Peer review of "Sustainable Operation Structures in the Cross-Border Enterprise Ecosystem under Tariff Fluctuations: A Systematic Evaluation"

_systems, doi:10.3390/systems10060211_

Round 1

Reviewer 1 Report

The abstract should be revised. The abstract of the article should include at least one sentence concerning the purpose of the article, scope of the research, research limitations, design, methodology, approach, findings, practical implications, originality and main conclusions.

Before marking the contributions (p. 2) primary research questions (problems) should be given.

Proposition 2 (p. 6) give 2 formula for calculating the optimal sales price and sales volume. It this with practical implication? How much time does a practitioner need to collect data and to make the calculations?

Assumption 1 (p. 5) "marginal revenues" and "marginal costs" are more theoretical than practical. I have seen them just in textbooks, not in business. If they have practical implication, please, give it.

The paper proposes too many and too complex calculations. They are time consuming when collecting data. The authors may rethink if the application of the proposed models is useful for business since their application is difficult and probably in some cases impossible.

Reviewer 2 Report

This article offers fresh insight into transnational businesses affected by tariff changes. The primary contribution of the authors is the modeling and simulation of the system dynamics of the described region. The subject matter is highly significant, but I have many concerns regarding the English language, formatting, and organization of the article.
I could list numerous errors in the article, but for me to accept it, it must meet the following criteria:
- Line 43 begins with "But" followed by "But little work [...]" and line 48 "But risk management [...]" indicates that the authors lack language terminology. This occurs repeatedly throughout the document.
- The additional materials are unreadable. Their organization discourages reading. Please correct this; Draw.io can be used to create some drawings. The drawings will be considerably more eye-catching. Please consider this tool in your future work.
Figures 1 and 2 require enhancement. Please remove the border from Figure 1 and enlarge it by 10%. The same applies to figure 2.
- Figure 3 is undersized. It would be preferable to divide it into two sections, such as vertically.
- There are missing spaces in the text. Example line 275 Please reread the text and correct the error.
- Adjust the gap between equation 7 and 8
- Adjust the numbers from 4 to 9 They are impossible to read. Simply put, they are too small.
- Appendices A, B, and C are in total disarray. Please correct it
I would like to compliment you on the conclusion. I believe the conclusion will be more valuable if you refer to the line 58 paragraph regarding the four contributions.

Author Response

Thank you for the comments on our paper. Please see the attachment. 

Round 2

Reviewer 2 Report

The authors greatly contributed to the research and corrected most of the issues that were addressed. I know that some things need improving (like better data visualization), but I believe that further research would be better. Thank you for kind response. I aprove the paper